# Impact of CD40 (rs1883832) and CD40L (rs1126535) gene variants on laryngeal cancer susceptibility and their association with serum biomarker levels of sCD40 and sCD40L

**Alper Gümüş** [1,2]*, **Dilara Sönmez**[3], **Şeyda Demirkol**[3], **Mehmet Tolgahan Hakan** [3], **Ayşegül Verim**[4], **Yusufhan Süoğlu**[5], **İlhan Yaylım**[3], **Arzu Ergen**[3]

1 Medical Biochemistry Laboratory, Çam Sakura City Hospital, University of Health Sciences, Istanbul, Turkey, 2 Institute of Health Sciences, Istanbul University, Istanbul, Turkey, 3 Department of Molecular Medicine, Aziz Sancar Institute of Experimental Medicine, Istanbul University, Istanbul, Turkey, 4 Department of Otorhinolaryngology/Head and Neck Surgery, Haydarpasa Numune Education and Research Hospital, Istanbul, Turkey, 5 Istanbul Faculty of Medicine, Department of Otorhinolaryngology, Istanbul University, Istanbul, Turkey

* dralpergumus@gmail.com

## Abstract

### Introduction

The most prevalent head and neck cancer type is laryngeal cancer. Laryngeal cancer susceptibility is increased by a combination of genetic variables and environmental factors. Genetic predispositions that influence the functioning of the immune system can affect tumor development. Our study investigates the impact of alterations in CD40 (rs1883832) and CD40L (rs1126535) genes and the levels of their proteins on the development of laryngeal cancer.

### Materials and methods

The PCR-RFLP method was used for genotyping SNPs in 96 patients with laryngeal cancer and 127 healthy individuals. Additionally, ELISA was utilized to measure circulating levels of sCD40 and sCD40L.

### Results

We identified a significant difference in the genotype distribution of CD40 (rs1883832) between laryngeal cancer patients and healthy individuals (p = 0.05). The C allele was dominant, and the CC genotype was more frequently observed in patients with laryngeal cancer (OR: 2.34, 95% CI: 0.98–5.54). In contrast, no statistically significant difference in the genotypes of CD40L (rs1126535) was detected between laryngeal cancer patients and the control group (p = 0.12). Additionally, no significant differences in serum sCD40 or sCD40L levels were observed between the groups (p = 0.48 and p = 0.15, respectively). However, a moderate positive correlation was found between sCD40 and sCD40L levels in the laryngeal cancer group (r = 0.52, p<0.01), a relationship that was not observed in the control group.

**Data Availability Statement:** All data are in the manuscript and/or supporting information files.

**Funding:** This work was funded by the Scientific Research Projects Coordination Unit of Istanbul University (Project no: TDK-2019-35062). The funders had no role in study design, data collection and analysis, decision to publish, or preparation of the manuscript.

**Competing interests:** The authors have declared that no competing interests exist.

**Abbreviations:** SNP, Single Nucleotide Polymorphism; CD40, Cluster of Differentiation 40 Protein; CD40L, Cluster of Differentiation 40 Ligand Protein; TNM, Malign Tumor Classification; PCR, Polymerase Chain Reaction; RFLP, Restriction Fragment Length Polymorphism; ELISA, Enzyme Linked Immune Sorbent Assay.

## Discussion

According to the current findings, it is suggested that the CD40 (rs1883832) gene variation found in patients may indicate an individual's susceptibility to developing laryngeal cancer. On the other hand, CD40L (rs1126535) seems to not play a significant role. While serum sCD40 and sCD40L levels did not show significant differences between patients and controls, the correlation in cancer patients suggests that these markers may be relevant in tumor progression. Further research is required to clarify the functional implications of these genetic variants and their potential use as biomarkers for laryngeal cancer.

## 1 Introduction

Laryngeal cancer makes up one-third of head and neck cancers, which are the sixth most prevalent cancers globally. They occur in different sites of the larynx, each with distinct symptoms and treatments. More than 90% of these cancers are classified as squamous cell type, which means they begin in the thin, flat cells that line the larynx. Less common adenocarcinoma starts in the glandular cells of the larynx [1,2]. The reported incidence rate in Turkey is 5.7% for men and 0.4% for women [3]. Various risk factors are associated with laryngeal cancer, such as smoking, excessive alcohol consumption, exposure to specific chemicals and pollutants, and contracting HPV (human papillomavirus) infection. [4,5]. Although there are known environmental risk factors, not all individuals with these factors develop laryngeal cancer. Polymorphism is the occurrence of more than one form or allele of a particular gene in a population. Different alleles can cause variations in traits, such as physical characteristics and health status. Certain genetic polymorphisms have been linked to a higher chance of developing cancer. For example, specific single nucleotide polymorphisms (SNPs) are associated with an increased risk of certain types of cancer [6]. It has been reported that SNPs, which affect many processes such as DNA repair and apoptotic, inflammatory, and oxidative processes, increase susceptibility to laryngeal cancer [7,8]. The role of the immune system in the development and progression of cancer is complex and multifaceted. The immune system constantly monitors the body for abnormal cells, such as cancer cells. The immune surveillance process is believed to have a crucial role in detecting and removing cancer cells at an early stage. In addition, cancer cells have devised various methods to avoid the immune system and hinder its capacity to combat them. Tumor-associated macrophages, myeloid-derived suppressor cells, and regulatory T cells are among the cells that can suppress the immune response [9,10]. The CD40/CD40L axis plays a significant role in cancer development. The interaction between CD40 and CD40L can promote cell proliferation, supporting tumor growth. Under certain conditions, it can also induce apoptosis, leading to the death of tumor cells. In addition, this axis activates immune cells, enhancing the immune response within the tumor microenvironment. This prevents tumor cells from escaping immune surveillance, facilitating their recognition and elimination by the immune system [11]. Some tumors may facilitate the development of tumors by producing molecules to suppress the immune system. CD40 is a protein involved in communication between cells on the cell surface. CD40 is essential for the stimulation of T cells [12]. Elevated levels of CD40 have been reported in some types of cancer [13–15]. CD40-producing cells simultaneously synthesize CD40L receptors. Similarly, it has been reported that CD40L levels are increased in cancer patients [16–19]. The CD40 (rs1883832) polymorphism located on chromosome 20 has been linked to an increased vulnerability to certain types of cancer [20,21]. It is thought that the effect of susceptibility to cancer is made by

altering the level of CD40 protein. It has been reported that carrying the C allele reduces the translation of the CD40 gene [19]. The gene encoding CD40L is located in the Xq26 region on the X chromosome and consists of five exons and four introns [22]. Our literature review did not locate a study evaluating whether CD40 (rs1883832) and CD40L (rs1126535) polymorphisms increase susceptibility to laryngeal cancer. Based on this information, our aimed to assess the potential impacts of CD40 (rs1883832) and CD40L (rs1126535) gene polymorphisms on the susceptibility to larynx cancer as well as serum levels of sCD40 and sCD40L proteins to determine their association with larynx cancer.

## 2 Materials and methods

### 2.1. Sample collection

This study was conducted with the approval of the Ethical Committee of the Istanbul Faculty of Medicine, Istanbul University (939/2019) according to the Declaration of Helsinki. The study was designed prospectively, and the samples were collected between 2020 and 2021. All participants provided written informed consent prior to inclusion in the study. A total of 96 patients with a confirmed diagnosis of laryngeal cancer and 127 healthy volunteers were initially recruited. Patients with a pathological diagnosis of laryngeal cancer were enrolled without restrictions on disease duration, severity, or treatment history. All participants were over the age of 18. Individuals with other chronic or acute inflammatory conditions were excluded from the study. Based on the exclusion criteria, eight patients were deemed ineligible and subsequently excluded, including data from 88 patients in the final analysis. Both the patient and control groups consisted exclusively of male participants. Detailed information regarding age, gender, time of diagnosis, smoking history, and the presence of any other systemic diseases was collected from all participants through direct interviews. The clinicopathological data of the laryngeal cancer patients were systematically collected. Two milliliters of whole blood were drawn from both patients and healthy controls into $K_2$EDTA additive tubes. Additionally, 5 mL of blood was collected in gel separator tubes to obtain serum samples. No pre-processing was performed on the whole blood samples. However, serum samples were processed by centrifugation at 1500 g for 10 minutes. Following collection, all samples were stored at -80˚C until further analysis.

### 2.2. DNA isolation and PCR-RFLP application

Genomic DNA extraction of all participants was performed using the salting-out technique from whole blood samples collected into the EDTA-additive tubes. Gene polymorphisms were analyzed by PCR using locus-specific primers and restriction fragment length polymorphism (RFLP) [23]. After enzymatic degradation, the eluted fractions were separated using agarose gel electrophoresis and visible under UV light.

The nucleotide sequence of the primer sequences used to amplify the region where the CD40 (rs1883832) polymorphism is observed is as follows:

Forward Primer (F): 5'-ACACAGCAAGATGCGTCCCTAAACT-3'

Reverse Primer (R): 5'-TCCTTCTCATTCCCCACTCCCAACT-3'

The thermal cycling protocol consisted of an initial denaturation step at 95˚C for 5 minutes, followed by 30 cycles of denaturation at 94˚C for 45 seconds, annealing at 55˚C for 45 seconds, extension at 72˚C for 45 seconds, and a final extension step at 72˚C for 5 minutes. The resulting PCR product has a length of 334 base pairs (bp).

The nucleotide sequence of the primer sequences used to amplify the region where the CD40L (r 126535) polymorphism is observed is as follows:

Forward Primer (F): F:5'-GGACTGCCCATCAGCATGAAA-3'

Reverse Primer (R): R:5′-TCCATCATTTGGGTAGAACCAACCT-3′

The primer sequences design and synthesis were conducted by our study group.

The thermal cycling protocol consisted of an initial denaturation step at 95°C for 5 minutes, followed by 30 cycles of denaturation at 94°C for 45 seconds, annealing at 62°C for 45 seconds, extension at 72°C for 45 seconds, and a final extension step at 72°C for 5 minutes. The resulting PCR product has a length of 285 bp.

The NcoI (5'-CCATGG-3') restriction enzyme (5'-CCATGG-3') and MmeI (5′-TCCRACN20/N18-3′) was used.

A commercial human sandwich (Thermo Fisher Invitrogen Waltham, Massachusetts, ABD) enzyme-linked immunosorbent assay (ELISA) kit was used to analyze circulating levels of sCD40 and sCD40L. The analysis was conducted as a single measurement. The within-run and between-run coefficients of variation (CVs) were sCD40 5.5% and 7% and sCD40L 4% and 6.8%, respectively. The reported linear measurement ranges for sCD40 and sCD40L were 7.8–500 pg/ml and 160–10000 ng/L, respectively.

### 2.3. Statistical analysis

The data collected in this study were analyzed using SPSS version 21 (IBM, New York, NY, USA). Descriptive statistics, including the mean, standard deviation (SD), minimum (min), maximum (max), and percentage values, were calculated and reported. The normality of the sCD40 and sCD40L levels was initially evaluated visually through histogram plots, followed by assessments of Kurtosis and Skewness coefficients after excluding outliers. Additionally, the Shapiro-Wilk test was employed to evaluate the distribution's normality further [24]. Since the data followed a normal distribution, parametric tests were applied. Independent sample t-tests were conducted for pairwise group comparisons, and one-way ANOVA was used to compare sCD40 and sCD40L levels between different gene variants. Post hoc pairwise comparisons were performed using Tukey's test to assess specific group differences. The distribution of sCD40 and sCD40L levels was visualized using box plot graphics representations stratified by both study groups and genotypes.

Categorical variables, including genotype and allele frequencies, were compared between the laryngeal cancer and control groups using chi-square ($\chi 2$) tests at a 95% confidence interval. The relative risk associated with genotypic variations was estimated by calculating odds ratios (OR) along with their respective confidence intervals (CI). Pearson correlation analysis was performed to explore potential correlations between the measured parameters, with scatter plots graphic generated to visually represent these correlations for each group independently. Statistical significance was determined with a p-value threshold set at 0.05. Power analysis was performed using G*Power 3.1 software.

## 3 Results

A comparison of demographic variables and sCD40 and sCd40L levels between groups are presented in Table 1. No significant difference was observed between the patient and control groups regarding demographic variables. For the sCD40 and sCD40L levels, which were observed to fit the normal distribution, no significant difference was observed between the control and patient groups in the statistical comparison (p = 0.48, p = 0.15). In the comparison made according to genotypes, a statistically significant difference was observed between the groups ($\chi 2$ = 3.8 p = 0.05). Relative risk depending on genotype: TT/CC Odds ratio: 2.34 (CI 95%: 0.98–5.54), CT/CC Odds ratio: 1.55 (CI 95%: 0.85–2.83), TT+CT/CC Odds ratio:1.71 (CI 95%: 0.97–3.01), there was a significant difference between the groups regarding allele distributions ($\chi 2$ = 4.33, p = 0.037). Allele-related relative risk: C/T Odds ratio: 0.67 (CI 95%: 0.44–0.97) (Table 2, Fig 1A and 1B).

**Table 1. Demographical parameters of study groups.**

|  | Control |  | Patent |  | P value |
|---|---|---|---|---|---|
| **Age** | Min | Max | Min | Max |  |
|  | 36 | 87 | 42 | 89 | 0.12 |
|  | Mean | SD | Mean | SD |  |
|  | 57.6 | 11.7 | 60.6 | 9.3 |  |
| **Tobacco** | Smoking | Non-smoking | Smoking | Non-smoking |  |
|  | %0 | %100 | %98 (or 60 packets/ per year | %2 |  |
| **Diagnosis Age** | - | - | Mean | SD |  |
|  |  |  | 57.3 | 9.1 |  |
| **sCD40L (ng/L)** | Min | Max | Min | Max |  |
|  | 282.7 | 872 | 280 | 2412 | 0.48 |
|  | Mean | SD/SE | Mean | SD/SE |  |
|  | 535.9 | 157.2/24.5 | 683 | 538/97 |  |
| **sCD40 (pg/L)** | Min | Max | Min | Max |  |
|  | 2.16 | 200.1 | 4.9 | 91.4 | 0.15 |
|  | Mean | SD/SE | Mean | SD/SE |  |
|  | 49.8 | 337.7/5.3 | 48.7 | 18.4/3.1 |  |
| **T Score** |  |  | T Score | N |  |
|  |  |  | T0 | 0 |  |
|  |  |  | T1 | 9 |  |
|  |  |  | T2 | 12 |  |
|  |  |  | T3 | 60 |  |
|  |  |  | T4 | 14 |  |
| **N Score** |  |  | N Score | N |  |
|  |  |  | N1 | 53 |  |
|  |  |  | N2 | 37 |  |
|  |  |  | N3 | 3 |  |
|  |  |  | N4 | 2 |  |
| **M Score** |  |  | M Score |  |  |
|  |  |  | M0 | 88 |  |
|  |  |  | M1 | 0 |  |

(Min: Minimum, Max: Maximum, SD: Standard Deviation, SE: Standard Error, p values).

In this study, we did not observe a significant difference in CD40L (rs1126535). No statistically significant difference was observed between the groups ($\chi 2 = 4.21$, p = 0.12). Relative risk depending on genotype: TT/CC Odds ratio: 1.71 (CI 95%: 0.83–3.52) CT/CC Odds ratio: 1.55

**Table 2. Distribution of CD40 (rs1883832) gene variants in study groups.**

| CD40 (rs1883832) | Control (n/%) |  | Patient (n/%) |  | P value |
|---|---|---|---|---|---|
| **CC** | 39 | 24% | 38 | 43% | **0.05**[*] |
| **CT** | 64 | 51% | 40 | 45% |  |
| **TT** | 24 | 25% | 10 | 12% |  |
| **C allele** | 142 | 55% | 116 | 45% | **0.037**[*] |
| **T allele** | 112 | 65% | 60 | 35% |  |

[*] Statistically significant.

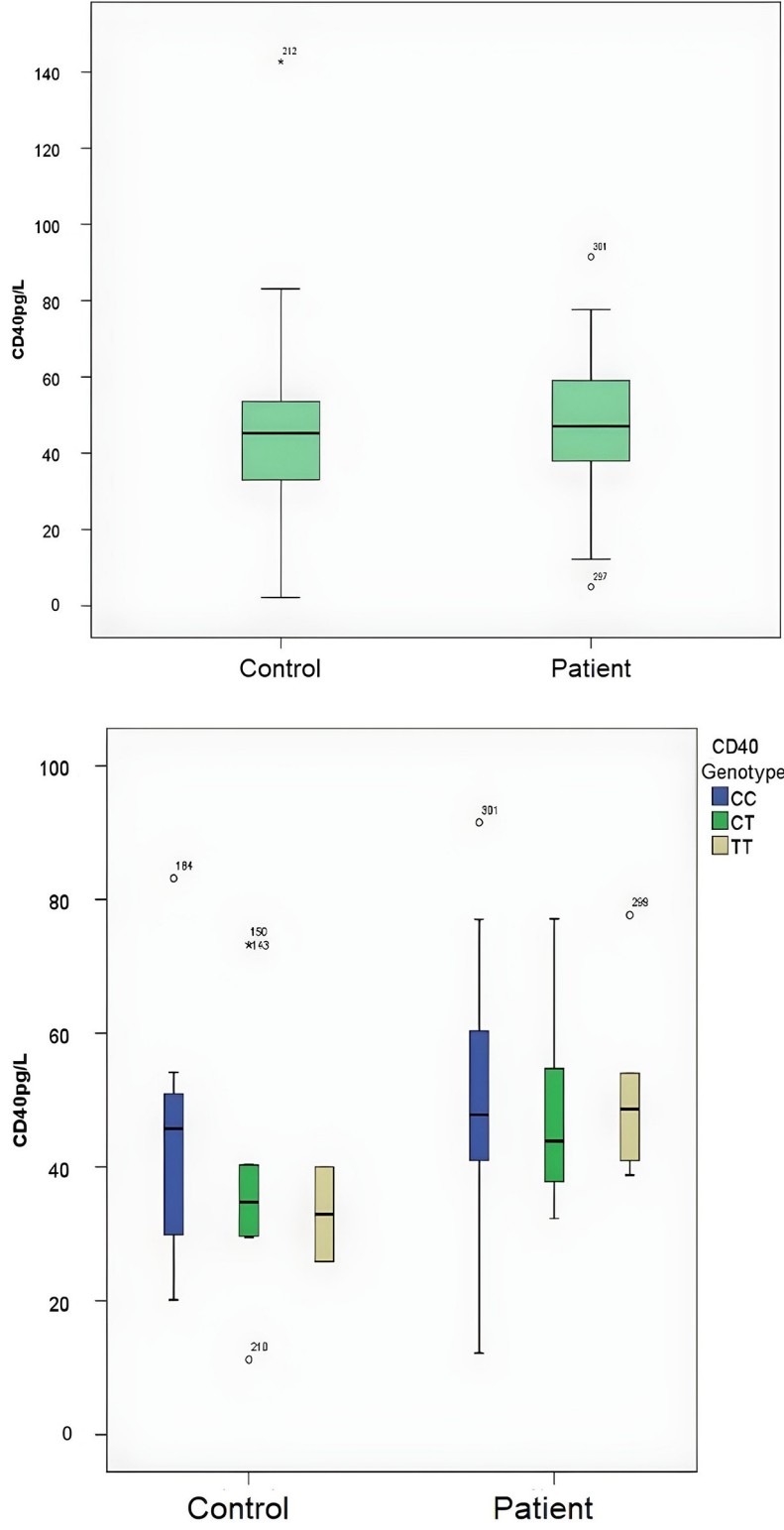

**Fig 1.** a. The prevalence of sCD40 levels by groups. b. Demonstration of sCD40 levels divided into subgroups according to genotypes. (*p = 0.037*).

**Table 3. Distribution of CD40 (rs1126535) gene variants in study groups.**

| CD40L (rs1126535) | Control (n/%) | | Patient (n/%) | | P value |
|---|---|---|---|---|---|
| TT | 47 | 37% | 31 | 35% | 0.12 |
| CT | 57 | 49% | 31 | 35% | |
| CC | 23 | 14% | 26 | 30% | |
| C allele | 103 | 40% | 83 | 47% | 0.17 |
| T allele | 151 | 60% | 93 | 53% | |

(CI 95%: 0.851.02–4.23), TT+CT/CC Odds ratio:1.71 (CI95%:1.09–3.9). There was no significant difference between the groups regarding alleles ($\chi2$ = 1.84, p = 0.17). Allele-related relative risk: C/T Odds ratio: 1.3 (CI 95%: 0.88–1.92) (Table 3, Fig 2A and 2B) polymorphism between patients with laryngeal cancer and the control group according to genotype or allele distribution (p = 0.12).

No significant difference was found in the clinical features of the patient group based on genotypes for both CD40 (rs1883832) and CD40L (rs1126535) when compared, including tumor stage and lymph node involvement (Tables 4 and 5). When the relationship between sCD40 and sCD40L was evaluated, no correlation was observed in the control group (r = 0.039, p = 0.89), while a significant positive correlation was observed in the patient group (r = 0.52, p< 0.01) (Fig 3A and 3B).

A power analysis was conducted based on the final sample sizes of 88 patients with laryngeal cancer and 127 healthy controls. Assuming a medium effect size (Cohen's d = 0.5), a significance level ($\alpha$) of 0.05, and a sample size ratio between groups, the power of the study was calculated to be 0.948.

## 4 Discussions

Although tobacco use, alcohol consumption, and viral infections are well-established environmental risk factors for laryngeal cancer, not all individuals exposed to these factors develop the disease. This indicates that genetic predispositions significantly impact individual susceptibility. Single nucleotide polymorphisms (SNPs) resulting from point mutations have been implicated in increasing cancer susceptibility [6]. SNPs are known to influence various cellular processes, including DNA repair, apoptosis, inflammation, and oxidative stress, all of which have been shown to contribute to the pathogenesis of laryngeal cancer [7,25]. The immune system is among the processes that are affected. CD40 is a protein involved in communication between cells on the cell surface and is essential for stimulating T cells [26,27]. It has been reported that CD40 levels are high in cancer types such as malignant melanoma, lung cancer, and stomach cancer [28–31]. CD40-producing cells simultaneously synthesize CD40L receptors. Similarly, it has been reported that CD40L levels are increased in cancer patients [16,17]. Stimulation of the CD40/CD40L axis destroys tumor cells by stimulating CD8 T lymphocytes with TNF-$\alpha$; conversely, it suppresses immunity by stimulating TNF-$\beta$ production and causing a tumorigenic effect [32]. At this point, it is thought that the situation that disrupts the balance is the continuous stimulation of the CD40 pathway [33]. After conducting a thorough literature review, we discovered that there is only one study that evaluates both sCD40 and sCD40L levels in patients with laryngeal cancer. Other than that, we could not find any other literature on this subject. According to Kaskas et al., there was no notable discrepancy in CD40L levels between individuals with head and neck cancer and those in the control group [34]. Similarly, we did not detect a significant difference in the levels of sCD40 and sCD40L between the control and patient groups, which was somewhat unexpected. Although no prior studies have

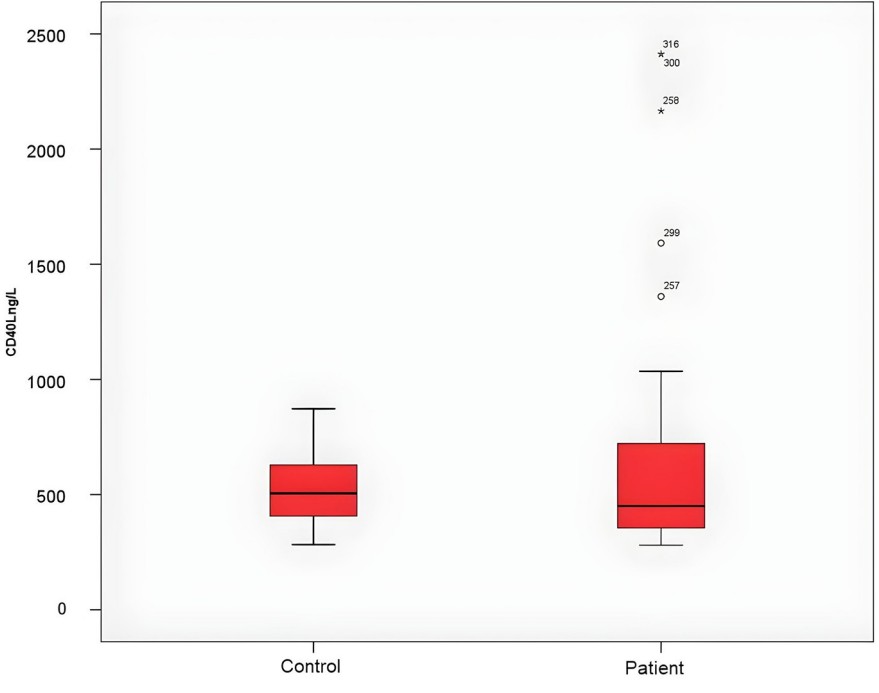

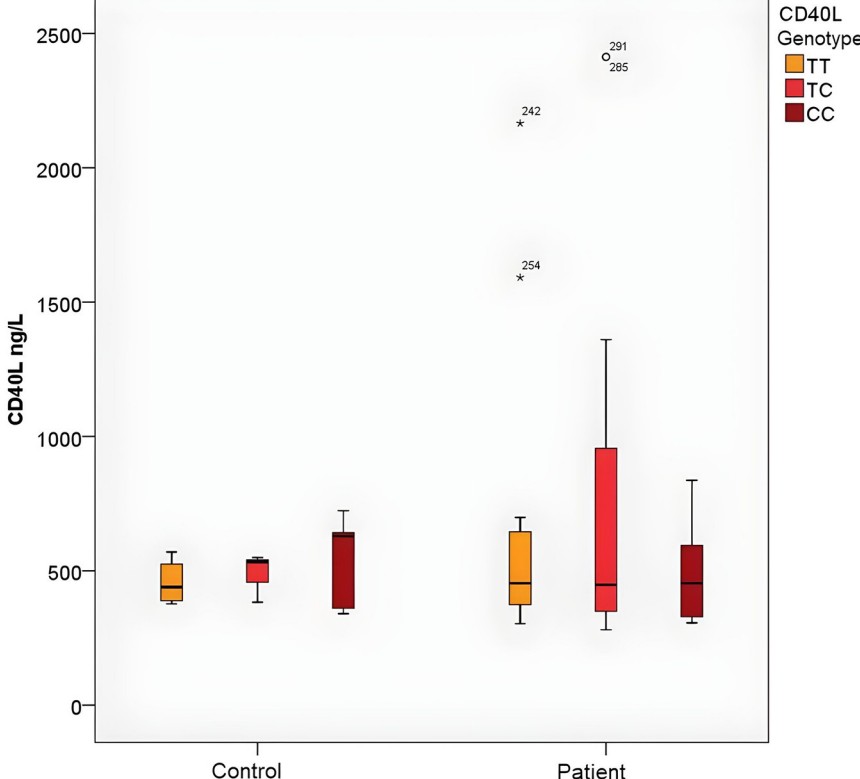

**Fig 2.** a. The prevalence of sCD40L levels by groups. b. Demonstration of sCD40 levels divided into subgroups according to genotypes. (p = 0.12).

**Table 4. The levels of sCD40 have been distributed based on tumor stage (T1+T2, T3+T4) or lymph node involvement (No, N1+N2+N3) with regard to CD40 (RS1883832).**

| Clinical feature | Min | Max | Mean | SD/SE | Min | Max | Mean | SD/SE | Min | Max | Mean | SD/SE | P Value |
|---|---|---|---|---|---|---|---|---|---|---|---|---|---|
| T1+T2 Value | 36.19 | 91.49 | 55.21 | 22.20/9.90 | 36.32 | 77.10 | 56.71 | 28.83/20.38 | 54.04 | 54.04 | 54.04 | - | 0.29 |
| T3+T4 Value | 12.19 | 77.00 | 48.244 | 20.71/7.32 | 32.29 | 77.10 | 47.10 | 13.25/3.99 | 38.79 | 77.65 | 51.52 | 17.92/8.96 | |
| N0 Value | 12.19 | 91.49 | 46.3 | 23.6/8.92 | 32.29 | 71.01 | 53.3 | 17.47/6.17 | 38.79 | 54.04 | 47.17 | 7.73/4.46 | 0.77 |
| N1+N2+N3 | 32.73 | 77 | 56.31 | 17.08/6.97 | 35.15 | 50.13 | 41.02 | 6.0/2.68 | 38.79 | 54.04 | 47.17 | 7.73/4.46 | |

(Min: Minimum, Max: Maximum, SD: Standard Deviation, SE: Standard Error).

specifically examined CD40 and CD40L levels in laryngeal cancer, research on other cancer types consistently reports an increase in CD40L levels alongside elevated CD40 levels in malignancies. To investigate this finding further, we explored the potential relationship between sCD40 and sCD40L levels. Although no correlation was observed in the control group, a significant association was identified in the patient group. Upon detailed examination of the clinical profiles of patients with elevated levels of both sCD40 and sCD40L, we found that these individuals were predominantly in the advanced stages of the disease. Supporting our findings, the study by Fujimoto et al. provides data that aligns with our conclusions [35]. It's essential to acknowledge that immune responses can change during the progression of cancer, which is a complex and extended process. However, the sCD40/sCD40L axis seems to become more active when the immune system is stimulated. Therefore, examining these two variables together may provide a more comprehensive understanding.

This study, conducted in Turkish society, found a difference between the laryngeal cancer patient group and the control group regarding CD40 (rs1883832) genotype and allele distributions. CD40 (rs1883832) polymorphism has been reported to increase susceptibility to lung cancer, Hodkin Lymphoma, and cervical cancer [20,26,36,37]. After conducting a thorough literature review, we could not find any studies that have specifically addressed the CD40 (rs1883832) polymorphism in patients with laryngeal cancer. It was reported that the CD40 polymorphism causes changes primarily in translation rather than transcription. Cancer susceptibility is believed to be exerted by altering the levels of the CD40 protein. It has been reported that carrying the C allele reduces the translation of the CD40 gene [38]. On the other hand, no differences have been detected in CD40 mRNA levels [39]. It is thought that the CD40 (rs1883832) polymorphism affects the initiation capacity of the ribosome in translation but does not affect the transcription capacity of RNA polymerase [40]. However, our study did not observe a significant difference in sCD40 levels between the laryngeal cancer patient group and the control group. Similarly, there was no significant difference in sCD40 levels among

**Table 5. The levels of sCD40L are presented based on tumor stage (T1+T2, T3+T4) or lymph node involvement (No, N1+N2+N3) in relation to the CD40L (RS1126535) polymorphism.**

| CD40L (ng/L) | CC | | | | CT | | | | TT | | | | |
|---|---|---|---|---|---|---|---|---|---|---|---|---|---|
| Clinical feature | Min | Max | Mean | SD/SE | Min | Max | Mean | SD/SE | Min | Max | Mean | SD/SE | P Value |
| T1+T2 Value | 303 | 303 | 303 | - | 280 | 2412 | 848 | 1043/521 | 328 | 455 | 411 | 72/41 | 0.74 |
| T3+T4 Value | 379 | 379 | 379 | - | 282 | 2412 | 788 | 637/192 | 306 | 836 | 532 | 235/106 | |
| N0 Value | 376 | 570 | 456 | 87/43 | 331 | 2412 | 964 | 394/149 | 306 | 836 | 452 | 182/68 | 0.70 |
| N1+N2+N3 | 351 | 2165 | 706 | 717/292 | 280 | 1360 | 621 | 394/149 | 728 | 728 | 728 | - | |

(Min: Minimum, Max: Maximum, SD: Standard Deviation, SE: Standard Error).

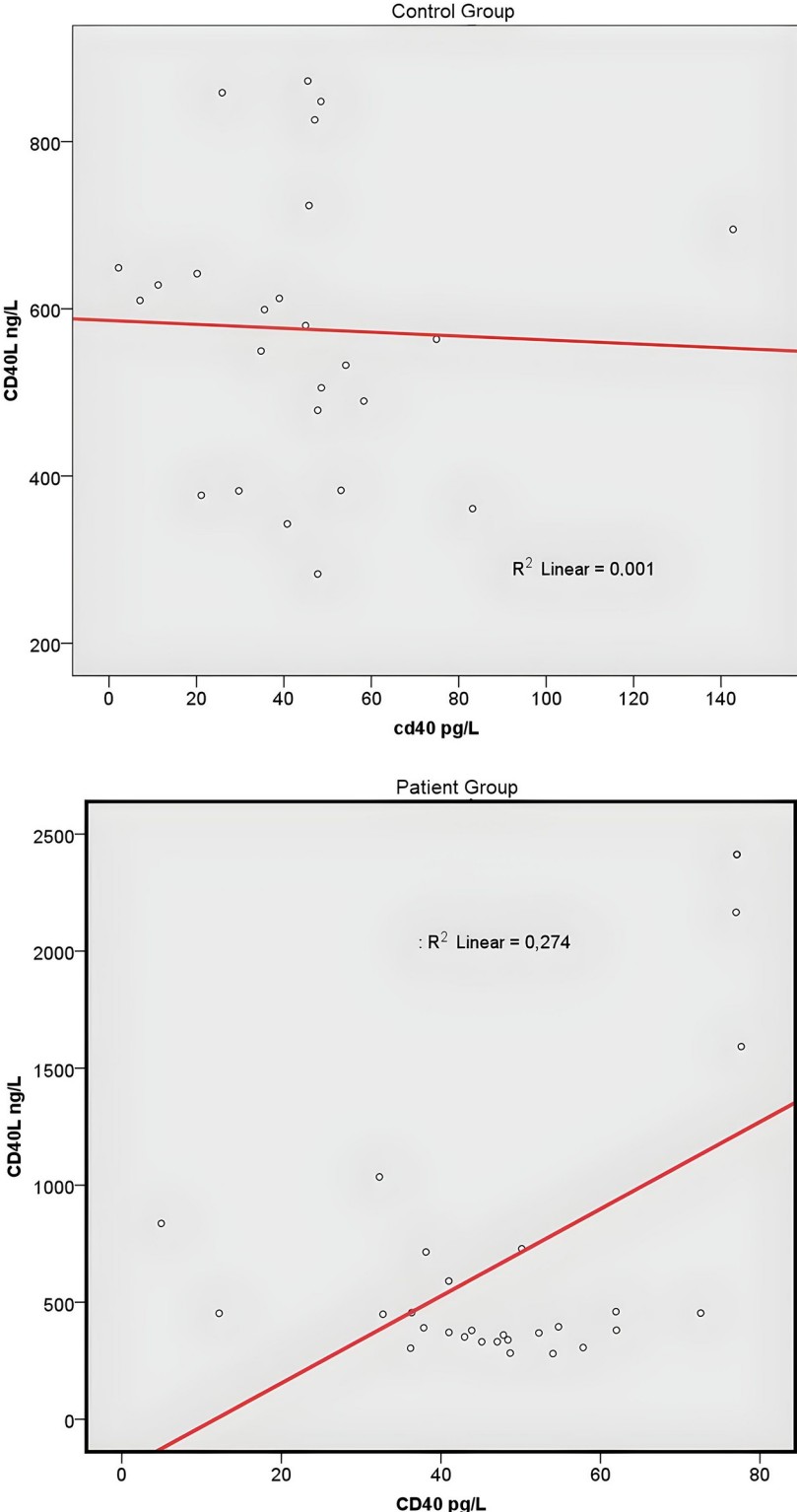

**Fig 3.** a. Scatter diagram of sCD40 and sCD40L values in the control group. b. Scatter diagram of sCD40 and sCD40L values in the patient group.

subgroups created based on clinical variables such as cancer stage or lymph node metastasis in the patient group. In attempting to understand why there are differences in CD40 genotypes but not in sCD40 levels, we believe that the impact of the CD40 (rs1883832) polymorphism on laryngeal cancer may be more closely tied to changes in the structure and function of the protein that is produced, rather than in the levels of gene products. Although it has been reported that the CD40 (rs1883832) polymorphism is located in the promoter region, we could not find any studies in the literature that investigate the functional changes associated with polymorphic CD40s.

Immunotherapeutic interventions could represent an alternative approach to evaluate the CD40/CD40L axis, given the multifaceted effects of CD40 signaling, which position it as a potential target in cancer immunotherapy. Specifically, targeting CD40 and CD40L through inhibitory or agonistic strategies offers promising outcomes by modulating the immune response. Therapies focused on this axis have shown clinical efficacy, suggesting new opportunities in cancer treatment. However, the success of such therapies is not without challenges. Straska et al. reported that monotherapy with an anti-CD40 agonist yielded minimal or no clinical response in patients with head and neck cancer, suggesting that combination therapies may be more efficacious [41]. To induce a robust antitumor effect, agonistic CD40 antibodies can modulate T cells, primarily via dendritic cells. Nonetheless, as Salomon et al. have noted, human CD40 antibodies exhibit limited antitumor activity, with low efficacy and dose-limiting toxicity observed in cancer patients [42]. In line with these findings, our data, showing no significant difference in sCD40 levels, further supports the notion that monotherapy targeting this pathway may not yield sufficient therapeutic benefit.

Our study did not reveal a significant difference in the CD40L (rs1126535) polymorphism between laryngeal cancer patients and the control group regarding genotype or allele distribution. Furthermore, no significant differences were observed between the laryngeal cancer patients and healthy individuals when assessing the relationship between the CD40L (rs1126535) polymorphism and sCD40L levels. Additionally, after stratifying the laryngeal cancer patients based on their clinical characteristics, we found no significant variation in sCD40L levels concerning tumor stage or lymph node involvement. These findings suggest that the CD40L (rs1126535) polymorphism and sCD40L levels may not play a major role in the progression of laryngeal cancer, at least within the context of this study.

One of the limitations of our study is the exclusive inclusion of male participants. Since the patient group was entirely male, we also selected the control group from a male population to ensure comparability. Unfortunately, no female patients were included during the study period. This limitation restricts our ability to assess potential immune system variations that may be influenced by hormonal factors, thereby limiting the generalizability of our findings across both genders.

Based on our research, the CD40 (rs1883832) polymorphism we detected in patients can be used as a criterion for determining the susceptibility of individuals to laryngeal cancer. In literature, some studies support our findings and those that contradict them. Inconsistent results may be due to many reasons, such as sample size, ethnic variables, and study design exclusion criteria. Together, we intend to expand and advance our study based on the information we acquired from our study. The topic demands more in-depth research. The literature needs more data on this subject and needs to be expanded with new and comprehensive studies. Also, we will design a longitudinal study to observe the progression of laryngeal cancer based on genetic markers.

## Supporting information

**S1 Raw data.**
(XLS)

## Acknowledgments

This study has been presented as an oral presentation at the MOKAD 2023, Istanbul. (https://avesis.istanbul.edu.tr/yayin/5a8d5607-cc03-4210-a345-9bb1eb0704cf/investigation-of-cd40-cd40l-gene-variants-and-scd40-scd40l-serum-levels-in-laryngeal-cancer)

We acknowledge the assistance of the GPT-4 AI language model by OpenAI for support with statistical analysis and drafting of this manuscript. All critical decisions and final revisions were performed by the human authors. No AI tool is credited as an author, adhering to submission guidelines.

## Author Contributions

**Conceptualization:** Alper Gümüş, Şeyda Demirkol, Ayşegül Verim, Yusufhan Süoğlu.

**Data curation:** Alper Gümüş, Dilara Sönmez, Mehmet Tolgahan Hakan, Ayşegül Verim, Yusufhan Süoğlu, Arzu Ergen.

**Formal analysis:** Alper Gümüş, Mehmet Tolgahan Hakan, Ayşegül Verim.

**Investigation:** Alper Gümüş, Dilara Sönmez, Şeyda Demirkol, Mehmet Tolgahan Hakan, Ayşegül Verim, Yusufhan Süoğlu, Arzu Ergen.

**Methodology:** Alper Gümüş, Dilara Sönmez, İlhan Yaylım, Arzu Ergen.

**Project administration:** Alper Gümüş, Mehmet Tolgahan Hakan, İlhan Yaylım, Arzu Ergen.

**Resources:** Şeyda Demirkol.

**Software:** Şeyda Demirkol, Mehmet Tolgahan Hakan.

**Supervision:** İlhan Yaylım, Arzu Ergen.

**Validation:** Dilara Sönmez, İlhan Yaylım, Arzu Ergen.

**Writing – original draft:** Alper Gümüş.

**Writing – review & editing:** Alper Gümüş.

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
