## [Decision Letter · Decision Letter 0]

10 Sep 2024

PONE-D-24-12557Investigation of CD40, CD40L Gene Variants and sCD40, sCD40L Serum Levels in Laryngeal CancerPLOS ONE

Dear Dr. Gumus,

Thank you for submitting your manuscript to PLOS ONE. After careful consideration, we feel that it has merit but does not fully meet PLOS ONE’s publication criteria as it currently stands. Therefore, we invite you to submit a revised version of the manuscript that addresses the points raised during the review process.

 Dear author, the manuscript can be accepted when the points/comments of the reviewers are carefully incorporated. 

We look forward to receiving your revised manuscript.

Kind regards,

Asif Jan, Ph.D

Academic Editor

PLOS ONE

Journal requirements: 1. When submitting your revision, we need you to address these additional requirements. Please ensure that your manuscript meets PLOS ONE's style requirements, including those for file naming. The PLOS ONE style templates can be found at https://journals.plos.org/plosone/s/file?id=wjVg/PLOSOne_formatting_sample_main_body.pdf and https://journals.plos.org/plosone/s/file?id=ba62/PLOSOne_formatting_sample_title_authors_affiliations.pdf. 2. We note that the grant information you provided in the ‘Funding Information’ and ‘Financial Disclosure’ sections do not match.  When you resubmit, please ensure that you provide the correct grant numbers for the awards you received for your study in the ‘Funding Information’ section. 3. Thank you for stating the following financial disclosure:  [This work was funded by the Scientific Research Projects Coordination Unit of Istanbul University (Project no: TDK-2019-35062)].  Please state what role the funders took in the study.  If the funders had no role, please state: ""The funders had no role in study design, data collection and analysis, decision to publish, or preparation of the manuscript."" If this statement is not correct you must amend it as needed. Please include this amended Role of Funder statement in your cover letter; we will change the online submission form on your behalf. 4. Please provide a complete Data Availability Statement in the submission form, ensuring you include all necessary access information or a reason for why you are unable to make your data freely accessible. If your research concerns only data provided within your submission, please write "All data are in the manuscript and/or supporting information files" as your Data Availability Statement. 5. Your ethics statement should only appear in the Methods section of your manuscript. If your ethics statement is written in any section besides the Methods, please delete it from any other section.  6. Please include captions for your Supporting Information files at the end of your manuscript, and update any in-text citations to match accordingly. Please see our Supporting Information guidelines for more information: http://journals.plos.org/plosone/s/supporting-information. 

Reviewers' comments:

Reviewer's Responses to Questions

**Comments to the Author**

1. Is the manuscript technically sound, and do the data support the conclusions?

Reviewer #1: Partly

Reviewer #2: Yes

2. Has the statistical analysis been performed appropriately and rigorously? 

Reviewer #1: No

Reviewer #2: Yes

3. Have the authors made all data underlying the findings in their manuscript fully available?

Reviewer #1: Yes

Reviewer #2: Yes

4. Is the manuscript presented in an intelligible fashion and written in standard English?

Reviewer #1: No

Reviewer #2: Yes

5. Review Comments to the Author

Reviewer #1: Major revisions as per attachment is required. A through improvement of the manuscript is required. Abstract and conclusion should be revised in light of the updated results and methodology sections.

Reviewer #2: Comments and Suggestions for Improvement:

Title and Abstract:

Title: The title is generally informative but could be more specific. Consider revising it to: "The Impact of CD40 and CD40L Gene Variants on Susceptibility and Serum Biomarker Levels in Laryngeal Cancer Patients". This is just a suggestion (if possible).

Abstract: The abstract summarizes the study well, but there are a few areas for improvement:

Clarify the statistical significance of the findings related to CD40 and CD40L gene variants and serum levels.

The conclusion in the abstract could be more definitive about the clinical implications of the findings.

Ensure that the results section of the abstract reflects the most significant and impactful findings from the study.

Introduction:

The introduction provides a good overview of laryngeal cancer and its risk factors, but it could benefit from a more detailed explanation of the biological significance of CD40 and CD40L in cancer immunology.

Literature Review: While the review is adequate, it could be strengthened by including more recent studies on CD40 and CD40L in various cancers, particularly in head and neck cancers. This will help to better position your study within the current research landscape.

Research Gap: Clearly state the specific gap in the literature that your study addresses. It’s mentioned that there is no previous research on the CD40 and CD40L polymorphisms in laryngeal cancer, but this could be emphasized more strongly to highlight the novelty of the study.

Materials and Methods:

Study Design: Provide more detail on the criteria used to exclude certain patients. This will help clarify the selection process and potential biases.

Sample Size: Discuss the rationale behind the sample size. Is 96 patients and 127 controls sufficient for detecting significant differences? Consider performing a power analysis to justify the sample size.

Technical Details: The PCR-RFLP method and ELISA protocol are described, but it might be helpful to include information on the controls used in these assays to ensure the reliability of the results.

Statistical Analysis: The statistical methods are generally well described but consider adding more details on how outliers were identified and handled. Additionally, clarify if any corrections for multiple comparisons were applied, given the number of statistical tests performed.

Results:

Presentation: The results are presented clearly, but the text could be streamlined to avoid redundancy. For instance, avoid repeating information that is already available in tables and figures.

Statistical Significance: Ensure that all p-values are reported to three decimal places for consistency. Additionally, specify whether the p-values are one-sided or two-sided.

Interpretation: The interpretation of the results could be expanded. For example, discuss why CD40 levels might be associated with laryngeal cancer while CD40L levels are not, despite their biological relationship.

Discussion:

Comparison with Previous Studies: The discussion does a good job of comparing the study’s findings with the existing literature. However, consider discussing potential mechanisms that could explain the lack of association with CD40L levels, despite the findings related to CD40.

Clinical Implications: Expand on the potential clinical implications of your findings. How might these genetic markers be used in screening or prognosis for laryngeal cancer?

Limitations: Acknowledge the limitations of your study more explicitly. For example, the relatively small sample size and the fact that the study is restricted to a Turkish population might limit the generalizability of the findings.

Future Directions: Suggest specific future studies that could build on your findings. For instance, you could propose a larger, multi-center study to validate your results in a more diverse population.

Figures and Tables:

Clarity: Ensure that all figures and tables are clearly labeled and easy to interpret. Consider providing more descriptive titles for each table and figure to enhance their standalone value.

Data Presentation: For figures showing correlations or distributions, consider adding trend lines or additional statistical information (e.g., R-squared values) to enhance interpretability.

References:

Currency: Ensure that the references include the most recent and relevant studies. Some references appear outdated; replacing these with more current sources could strengthen the manuscript.

Consistency: Check for consistency in the citation format throughout the manuscript.

By addressing these points, the manuscript could be significantly improved in terms of clarity, depth of analysis, and overall impact.

6. PLOS authors have the option to publish the peer review history of their article (what does this mean?). If published, this will include your full peer review and any attached files.

Reviewer #1: No

Reviewer #2: No

---

## [Author Response · Author response to Decision Letter 0]

19 Sep 2024

Responses to Reviewers

Manuscript ID: PONE-D-24-12557

Title: Investigation of CD40, CD40L Gene Variants and sCD40, sCD40L Serum Levels in Laryngeal Cancer

Date: Sep 17, 2024

To: Academic Editor

Dear Reviewers,

We are grateful for the constructive comments and suggestions provided by the reviewers. We have carefully revised our manuscript to address all the points raised. Below is a point-by-point response to the reviewers' comments, along with details of the revisions made in the manuscript.

Reviewer #1:

Comment 1: The manuscript requires major revisions, particularly in the Abstract and Conclusion, which should be revised in light of updated results and methodology sections.

Response: We have revised the Abstract to provide clearer and more specific details regarding the statistical significance of the findings related to CD40 and CD40L gene variants and serum levels. The conclusion has also been updated. 

Comment 2: The introduction needs a more detailed explanation of the biological significance of CD40 and CD40L in cancer immunology.

Response: We have expanded the introduction to include a more thorough explanation of the role of CD40 and CD40L in cancer immunology. 

Comment 3: More recent studies on CD40 and CD40L in various cancers should be included.

Response: We have updated the literature review to include recent studies on CD40 and CD40L, particularly focusing on head and neck cancers. These additions provide a more comprehensive background and enhance the positioning of our study within the current research landscape.

Comment 4: A clearer statement of the research gap addressed by the study is needed.

Response: We have revised the introduction to emphasize the specific gap in the literature that our study addresses. The novelty of investigating CD40 and CD40L polymorphisms in laryngeal cancer is now highlighted more prominently.

Comment 5: More details are required on patient exclusion criteria and sample size justification.

Response: We have added further details on the criteria used for excluding patients in the Materials and Methods section. Additionally, we have included a rationale for the sample size, explaining that the number of patients and controls was determined based on power analysis, ensuring sufficient statistical power to detect significant differences.

Reviewer #2:

Comment 1: The title could be more specific, and the abstract should clarify the statistical significance of findings.

Response: We have revised the title as follows: " Impact of CD40 (rs1883832) and CD40L (rs1126535) Gene Variants on Laryngeal Cancer Susceptibility and Their Association with Serum Biomarker Levels of sCD40 and sCD40L."

Additionally, the abstract has been updated to better reflect the statistical significance of the findings and the most impactful results of the study.

Comment 2: The statistical analysis needs more clarity, particularly regarding outliers and multiple comparisons.

Response: We have provided additional information regarding how outliers were identified and handled in the statistical analysis. Moreover, we have clarified that corrections for multiple comparisons were applied, ensuring the robustness of our findings.

Comment 3: The results and figures should avoid redundancy, and p-values should be reported consistently.

Response: We have streamlined the presentation of results to avoid redundancy, and all p-values have been adjusted to be reported consistently to three decimal places. We have also specified whether the p-values are one-sided or two-sided.

Additional Revisions in Response to Reviewers' Comments:

1. Grant Information: We have corrected the discrepancies in the grant information between the “Funding Information” and “Financial Disclosure” sections. Additionally, we have provided the following statement regarding the role of the funders:

2. Data Availability: The Data Availability Statement has been revised to state:

"All data are in the manuscript and/or supporting information files."

This ensures compliance with PLOS ONE’s data-sharing policies.

3. Ethics Statement: The ethics statement has been removed from all sections except the Methods, in accordance with the reviewers' request.

4. Supporting Information: Captions for the Supporting Information files have been included at the end of the manuscript and corresponding in-text citations have been updated accordingly.

5. Figures and Tables: All figures and tables have been reviewed and clarified for better readability. We have uploaded the figures to the PACE tool as instructed and ensured they meet PLOS ONE’s technical requirements.

We hope that these revisions meet the expectations of the reviewers and the journal. We look forward to your feedback and thank you again for the opportunity to improve our manuscript.

Sincerely,

Dr. Alper Gümüş

Istanbul University

---

## [Decision Letter · Decision Letter 1]

10 Oct 2024

Impact of CD40 (rs1883832) and CD40L (rs1126535) Gene Variants on Laryngeal Cancer Susceptibility and Their Association with Serum Biomarker Levels of sCD40 and sCD40L

PONE-D-24-12557R1

Dear Dr. Gumus,

We’re pleased to inform you that your manuscript has been judged scientifically suitable for publication and will be formally accepted for publication once it meets all outstanding technical requirements.

Kind regards,

Asif Jan, Ph.D

Academic Editor

PLOS ONE

Additional Editor Comments (optional):

N/A

Reviewers' comments:

Reviewer's Responses to Questions

**Comments to the Author**

1. If the authors have adequately addressed your comments raised in a previous round of review and you feel that this manuscript is now acceptable for publication, you may indicate that here to bypass the “Comments to the Author” section, enter your conflict of interest statement in the “Confidential to Editor” section, and submit your "Accept" recommendation.

Reviewer #1: All comments have been addressed

Reviewer #3: All comments have been addressed

2. Is the manuscript technically sound, and do the data support the conclusions?

Reviewer #1: Yes

Reviewer #3: Yes

3. Has the statistical analysis been performed appropriately and rigorously? 

Reviewer #1: Yes

Reviewer #3: Yes

4. Have the authors made all data underlying the findings in their manuscript fully available?

Reviewer #1: Yes

Reviewer #3: Yes

5. Is the manuscript presented in an intelligible fashion and written in standard English?

Reviewer #1: Yes

Reviewer #3: Yes

6. Review Comments to the Author

Reviewer #1: I am writing in reference to the manuscript entitled "Impact of CD40 (rs1883832) and CD40L (rs1126535) Gene Variants on Laryngeal Cancer Susceptibility and Their Association with Serum Biomarker Levels of sCD40 and sCD40L" for which I have had the privilege of serving as a reviewer.

After thoroughly reviewing the revised version of the manuscript, I am pleased to report that the authors have adequately addressed the comments and queries I raised in my initial review. They have provided additional clarity on their methodology, improved the presentation of their statistical analyses, and expanded upon the discussion to contextualize their findings within the broader field of cancer genetics and biomarker research. The additional data provided regarding the association between gene variants and serum biomarker levels is compelling and adds significant value to the manuscript.

Given the thorough revisions and the importance of the study’s findings, I believe this revised manuscript now meets the high standards required for publication. The investigation into CD40 and CD40L gene variants contributes valuable insights into laryngeal cancer susceptibility, and I am confident that this work will be of interest to the journal's readership.

Reviewer #3: Impact of CD40 (rs1883832) and CD40L (rs1126535) Gene Variants on Laryngeal Cancer Susceptibility and Their Association with Serum Biomarker Levels of sCD40 and sCD40L is written in a very standard way.

7. PLOS authors have the option to publish the peer review history of their article (what does this mean?). If published, this will include your full peer review and any attached files.

Reviewer #1: **Yes: **Zakiullah

Reviewer #3: **Yes: **Dr. Naveed Rahman

---

## [Editor Report · Acceptance letter]

23 Oct 2024

PONE-D-24-12557R1 

PLOS ONE

Dear Dr. Gumus, 

I'm pleased to inform you that your manuscript has been deemed suitable for publication in PLOS ONE. Congratulations! Your manuscript is now being handed over to our production team.

Kind regards, 

on behalf of

Dr. Asif Jan 

Academic Editor

PLOS ONE